# Direct Polyphenol Attachment on the Surfaces of Magnetite Nanoparticles, Using *Vitis vinifera*, *Vaccinium corymbosum*, or *Punica granatum*

**DOI:** 10.3390/nano13172450

**Published:** 2023-08-30

**Authors:** Ana E. Matías-Reyes, Margarita L. Alvarado-Noguez, Mario Pérez-González, Mauricio D. Carbajal-Tinoco, Elizabeth Estrada-Muñiz, Jesús A. Fuentes-García, Libia Vega-Loyo, Sergio A. Tomás, Gerardo F. Goya, Jaime Santoyo-Salazar

**Affiliations:** 1Departamento de Física, Centro de Investigación y de Estudios Avanzados del Instituto Politécnico Nacional, CINVESTAV-IPN, Mexico City 07360, Mexico; margarita.alvarado@cinvestav.mx (M.L.A.-N.); mauricio.carbajal@cinvestav.mx (M.D.C.-T.);; 2Área Académica de Matemáticas y Física, Instituto de Ciencias Básicas e Ingeniería, Universidad Autónoma del Estado de Hidalgo, UAEH, Mineral de la Reforma 42184, Mexico; mario_perez12865@uaeh.edu.mx; 3Departamento de Toxicología, Centro de Investigación y de Estudios Avanzados-IPN, Av. IPN No. 2508, Col. San Pedro Zacatenco, Ciudad de México 07360, Mexicolvega@cinvestav.mx (L.V.-L.); 4Instituto de Nanociencia y Materiales de Aragón (INMA), CSIC-Universidad de Zaragoza, Campus Río Ebro, 50018 Zaragoza, Spain; 5Departamento de Física de la Materia Condensada, Facultad de Ciencias, Universidad de Zaragoza, 50009 Zaragoza, Spain

**Keywords:** polyphenols, natural products, magnetic nanoparticles, inductive heating, magnetic domains, cell viability

## Abstract

This study presents an alternative approach to directly synthesizing magnetite nanoparticles (MNPs) in the presence of *Vitis vinifera*, *Vaccinium corymbosum*, and *Punica granatum* derived from natural sources (grapes, blueberries, and pomegranates, respectively). A modified co-precipitation method that combines phytochemical techniques was developed to produce semispherical MNPs that range in size from 7.7 to 8.8 nm and are coated with a ~1.5 nm thick layer of polyphenols. The observed structure, composition, and surface properties of the MNPs@polyphenols demonstrated the dual functionality of the phenolic groups as both reducing agents and capping molecules that are bonding with Fe ions on the surfaces of the MNPs via –OH groups. Magnetic force microscopy images revealed the uniaxial orientation of single magnetic domains (SMDs) associated with the inverse spinel structure of the magnetite (Fe_3_O_4_). The samples’ inductive heating (H_0_ = 28.9 kA/m, f = 764 kHz), measured via the specific loss power (SLP) of the samples, yielded values of up to 187.2 W/g and showed the influence of the average particle size. A cell viability assessment was conducted via the MTT and NRu tests to estimate the metabolic and lysosomal activities of the MNPs@polyphenols in K562 (chronic myelogenous leukemia, ATCC) cells.

## 1. Introduction

Magnetite (Fe_3_O_4_) nanoparticles (MNPs) are the most promising superparamagnetic iron oxide nanoparticles (SPIONs) for use in specific functions such as drug release, diagnosis, and therapy. Their addressable single domains, particle size (<30 nm) [1], shape [2], composition [3], and surface properties [4] have been the key to tailoring the magnetic triggering of physicochemical functions in biological systems [5,6]. Theranostic core–shell formulations of SPIONs have been configured via direct attachment due to the affinity of the organic layers on their surfaces. The hydroxyl groups establish covalent bonds with these surfaces, and additional stability on MNPs as Fe–OH. The main functional groups that can react with the surfaces of SPIONs are phosphonic acid, carboxylic acid, catechol, hydroxyl, amine, glycerol, and siloxanes [7,8]. These organic grafts increase the possibility of the transference of the encapsulated SPIONs into aqueous media to achieve precise control over the nanoparticles’ performance in the cellular environment [9]. The green biosynthesis and wet-chemistry methods as coprecipitation and sonochemical synthesis, with some modifications, have opened new horizons in the processing of superparamagnetic MNPs with polyphenols, which are aromatic compounds that bear one or more hydroxyl groups [10,11,12,13].

Direct polyphenol coatings on the surfaces of MNPs have adjunct therapeutic potential because their phytochemical features exert important biological functions like free-radical scavenging and antioxidant and anti-inflammatory effects [14,15]. The specific biological properties of polyphenols have been recognized due to their alteration and prevention of chronic diseases such as neurodegenerative and cardiovascular diseases and osteoporosis. Additionally, their components play important roles in enhancing natural health-promoting actions, such as anti-tumor, anti-hypertensive, anti-thrombotic, and anti-diabetic effects and α-glucosidase inhibitory activities [16].

The diverse structures of polyphenols include valuable compounds present in plants, fruits, seeds, cereals, seasonings, chocolate, tea, wine, oils, and marine organisms that can prevent damage from reactive oxygen species (ROS) through, for example, free-radical scavenging or binding iron to prevent the generation of these species [17]. The classification of polyphenols includes phenolic acids, tannins, flavonoids, stilbenes, and lignans [18]. Also, polyphenolic compounds have been classified depending on their hydroxyl link positions and chemical skeleton variations, such as their degrees of oxidation, hydroxylation, methylation, and glycosylation, their ability to bond with other molecules, their solubility, and their abilities to minimize degradation processes, reduce toxicity, and control the active absorption and biological responses of polyphenols [19,20]. The antioxidant activities of polyphenols and their conjugates can be determined by their subclasses, and a decreasing trend was found in the following order: flavonols > flavanones > flavones. Herein, the positions of the hydroxyl groups are critical in developing coordination bonds with metal cations as metal–polyphenol nanoparticles, composite crystallization, and gelation [21,22].

The direct nanoencapsulation of SPIONs with polyphenolic surfaces is an effective alternative, improving their biocompatibility and attachment to the cell membrane; these qualities are consistent with the different stages of an active uptake process, increasing their therapeutic possibilities [23]. Furthermore, the effects of SPION-polyphenol particles can be associated with biochemical and molecular mechanisms within the intra- and inter-cellular signalling pathways, such as regulating nuclear transcription factors and fat metabolism and modulating the synthesis of inflammatory mediators [24].

This work focuses on the direct synthesis of magnetite nanoparticles coated with polyphenols derived from natural sources, specifically, *Vitis vinifera* (grape), *Vaccinium corymbosum* (blueberry), and *Punica granatum* (pomegranate) (Figure 1). The novelty of this approach lies in the eco-friendly and cost-effective phytochemical methods used to produce MNPs@polyphenols which have a heating capacity for the local release of antioxidant agents via a remote magnetic stimulus and are suitable for therapeutic applications. The in vitro tests assessing the biocompatibility of these materials indicated low cytotoxicity levels compatible with their potential therapeutic applications.

### Description and Properties of Polyphenols

*Punica granatum* is fruit rich in phytochemicals, with a content of 249.4 mg/g [25], containing 124 different polyphenols. These include flavonoids (anthocyanins and derivatives, catechin, epicatechin, and quercetin), tannins (ellagitannins, punicalagin, punicalin, and pedunculagin), and phenolic acids. Punicalagin has the largest molecular weight among these polyphenols [26,27]. Phytochemicals, being non-toxic, have been extensively studied for their health benefits, including their antioxidant, anti-inflammatory, anti-microbial, anti-infective, antidiabetic, and cancer-preventive effects [28,29,30]. Studies have shown inhibitory effects on HepmG2 and Hela cells, as well as anti-cancer activities in breast, colon, breast 20, cervical, and lung cancer cell lines [31,32,33].

*Vaccinium corymbosum* is rich in polyphenols, including 27 distinct anthocyanins, anthocyanidins (delphinidin, cyaniding, petunidin, peonidin, and malvidin), flavonols (quercetin, isoquercetin, and rutin), flavan-3-ols oligomeric derivatives (procyanidine A2, procyanidine B1, and procyanidine B2), flavan-3-ols, hydroxycinnamic acids (catechin, epicatechin, and epicatechin gallate), gallic acid, and protocatechualdehyde [34]. Quercetin is the most abundant flavonoid [35].

The high concentration of polyphenols makes them effective at treating diseases by combating oxidative stress and inflammation [36,37], offering potent antioxidant, anti-inflammatory, hepatoprotective [38], and cancer-preventive effects [36]. Polyphenols also have neuroprotective and cardioprotective properties and antidiabetic effects. The inhibition of tumor cell proliferation has been studied in vitro in colon (HT-29 and HCT116), prostate (LNCaP), breast, cervix (HeLa), ovarian (A2780), skin (B16F10) [36], and mouth [39] cells.

*Vitis vinifera*, like other fruits, contain polyphenols, including anthocyanins, flavanols (catechin and epicatechin), flavonols (rutin, quercetin-3-o-glucoside, rihydroquercetin-3-O-rhamnoside, isorhamnetin-3-O-hexose, isorhamnetin, and myricetin), flavonoids (consisting of (+)-catechins, (-)-epicatechin, and procyanidin polymers), phenolic acids (hydroxybenzoic acids methyl, dihydroxybenzoic acid, 1-O-vanilloyl-β-D-glucose, p-coumaric acid-4-glucoside, and caffeic acid), and stilbenes (resveratrol, resveratrol-3-oglucoside, viniferin, scripusin A, and piceatannol) [40,41]. In experimental studies, these polyphenols have been shown to have pharmacological effects, such as providing skin protection, antioxidant activity, antibacterial activity, anticancer properties, anti-inflammatory effects, and antidiabetic properties, as well as cardioprotective and neuroprotective effects [41,42,43]. Particularly, resveratrol has been reported to have antioxidant, anti-inflammatory, anti-apoptotic, anticancer, and anti-carcinogenic properties [44]. Some of its functions include the modulation of mitochondrial function and redox biology and dynamics, which have been observed in both in vitro and in vivo experiments.

In Table 1, the most important polyphenols in *Vitis vinifera*, *Vaccinium corymbosum*, and *Punica granatum* are shown.

## 2. Materials and Methods

### 2.1. Materials Used to Synthesize the MNPs@Polyphenols

Sodium hydroxide (NaOH), ferrous chloride tetrahydrate (FeCl_2_·4H_4_O), and ferric chloride hexahydrate (FeCl_3_·6H_4_O) were bought from Sigma-Aldrich. *Vitis vinifera*, *Punica granatum*, and *Vaccinium corymbosum* fruits were purchased from a local market in Mexico City. All reagents were used directly without further purification.

### 2.2. Cell Viability Materials

Dimethyl sulfoxide (DMSO) (Cat. D2650; 100% purity), MTT and NR were obtained from Sigma Chemical Co. (St. Louis, MO, USA); RPMI-1640, FBS, nonessential amino acids (100 mM), L-glutamine, sodium pyruvate, and an antibiotic–antimycotic were obtained from Invitrogen-Gibco. Other reagents were obtained from J.T. Baker, Inc. (Deventer, the Netherlands) as indicated.

### 2.3. Preparation of the Aqueous Extract

Each natural fruit extract was prepared according to the following method for the series of three syntheses: (1) *Vitis vinifera*, (2) *Punica granatum,* and (3) *Vaccinium corymbosum*. The fresh fruits were washed with distilled water before they were used. A specific amount of each fruit (75.391 g of *Vitis vinifera*, 44.56 g of *Punica granatum*, or 15 g of *Vaccinium corymbosum*) was placed in a 500 mL beaker, and 250 mL of distilled water was added. The resulting solution was warmed at 60 °C under magnetic stirring for 20 min. The final mixture was then cooled down to obtain an aqueous extract, which was reserved at room temperature for later use [47,48,49,50].

### 2.4. MNPs@Polyphenol Synthesis

Since particles with superparamagnetic properties have been reported in the size range of 10–30 nm, during the synthesis, some conditions, such as the temperature (70–80 °C), pH (10–14), and the volume of the polyphenol were controlled to obtain particle sizes within this range. The synthesis was performed following the co-precipitation method [51], with the addition of a polyphenol extract to directly coat the surfaces of the MNPs via Fe-OH bonding. The main control condition in this process was to reach a pH value in the alkaline range of 10–14. The volume of the polyphenol extract could show variations depending on its nature. The volume used in the synthesis was 20 mL for the MNPs@Polyphenols. The phytochemical is considered eco-friendly, cheaper, and more efficient in the substitution of some organic bases for the synthesis of MNPs and their stabilization and contributes to their direct capping [52].

The starting solution was prepared in a beaker with 54 mL of distilled water and 10 mL of hydrochloric acid (HCl) at a purity of 37%; the solution was bubbled with nitrogen gas for 30 min. Subsequently, in two 100 mL beakers, 25 mL and 6.25 mL of the previous solution were added, as were the salts of FeCl_3_ (6.996 g) and FeCl_2_ (2.52 g), respectively. The solutions were stirred at 500 rpm for 30 min. Then, 2.5 mL of the FeCl_2_·4H_4_O solution and 10 mL of the FeCl_3_·6H_4_O solution were added to a three-neck beaker under a nitrogen atmosphere. The mixture remained under constant stirring at 150 rpm and 70–80 °C for 15 min. Separately, a NaOH solution (7M) was prepared in 20 mL of deionized water and placed on a magnetic stirrer at 500 rpm for 5 min. After, the NaOH solution was added by dripping it into the three-neck beaker to obtain a pH of 12, and the solution turned a dark brown color. Afterwards, the solution was maintained under stirring for 5 min, and 20 mL of the respective aqueous extract was added to the mixture dropwise. The final black mixture was stirred for 20 min. It was then allowed to cool down to room temperature, and the nanoparticles were acquired via magnetic decantation.

The obtained black precipitates were washed with a mixture of deionized water and ethanol (in equal parts), sonicated for 5 min, and then allowed to precipitate via decantation in order to remove the excess organic base. This procedure was repeated three times, and then the MNPs@polyphenols products were washed twice with deionized water only. They were then washed to reach a pH of 7. Finally, the black precipitates were lyophilized, and each obtained powder sample was stored under a vacuum for further analysis.

## 3. Formation Mechanism of MNPs@Polyphenols

Polyphenols are phytochemicals with antioxidant/reducing attributes that allow them to mediate the reduction of various compounds into their corresponding nanoparticles. These biomolecules also contain a series of water-soluble metabolites. The use of phytochemicals as recovery/stabilization agents could reduce side effects and increase the biocompatibility of inorganic nanoparticles for biomedical applications.

The conditions of the reaction, such as the temperature, iron precursor concentration, amount of plant extract, reaction duration, type of polyphenol extract molecules, and pH, influence the physicochemical properties of the nanoparticle’s surface [53].

Fe_3_O_4_ nanoparticles were formed from a (Fe^3+^: Fe^2+^) 2:1 molar ratio of ferric hydroxide and ferrous hydroxide salts in an aqueous solution. The reaction Fe^2+^+ 2Fe^3+^+ 8OH^−^ → Fe_3_O_4_ + 4H_2_O confined the formation of the nanoparticles to saturation conditions with a pH of 10–14 [54]. In this work, the pH conditions were 13–14. For the coating process, the polyphenols contain the hydroxyl −OH in their elemental configuration, and these hydroxyls are covalently coupled to iron ions. This is achieved via the chelation of Fe^3+^−OH and Fe^2+^−OH on the surfaces of the Fe_3_O_4_ nanoparticles, and the polyphenols interact with the hydrogens of the -OH groups on the surfaces of the metal oxide nanoparticles. This interaction depends on the number of dents, or coordination sites, their functional groups possess [55]. Also, Wang et al. [56] suggested that the chelation of Fe_3_O_4_ with Sage (*Salvia officinalis*) mediated iron–polyphenol complex nanoparticles.
(1)$$Polyphenol+{H_{2}O}_{\left(L\right)}+2{Fe}^{3+}+{Fe}^{2+}\overset{stirring}{\to }\left[\frac{Polyphenol}{2{Fe}^{3+}{:Fe}^{2+}}\right]$$
(2)$$\left[\frac{Polyphenol}{2{Fe}^{3+}{:Fe}^{2+}}\right]+8OH_{\left(aq\right)}^{-}\to \frac{Polyphenol}{{Fe}_{3}O_{4}}+4H_{2}O$$

Also, this is an alternative strategy to add an available –OH linker on top of the surface of the Fe_3_O_4_ nanostructure. Considering that the MNPs can act as carriers, the polyphenol properties contribute to the additional formulations as nanocarriers.

## 4. Physicochemical Characterization

XRD measurements were performed using a Siemens D5000 X-ray diffractometer, Munchen, Germany, equipped with a copper radiation source (Cu Kα) with wavelength of 1.5418 Å, operating at accelerating voltage of 35 kV and a current of 25 mA. The measurements were carried out on powder samples in the 2θ range from 20° to 75° at a step rate of 0.02°. Match! (3.6.1.115, 2018, Bonn, Germany) and Powder Cell (2.3, 2001, Germany) software were used to analyze the XRD patterns. The diffractograms were compared with the COD Crystallography Open Database. The FTIR spectra of the nanoparticle powders were obtained via Nicolet 6700 FTIR equipment (Thermo Scientific, Waltham, MA, USA) operating between 3600 and 400 cm^−1^, with a resolution of 0.02 cm^−1^. Quantifications were conducted on the powder samples, and data were collected via OriginPro, Version 2022. OriginLab Corporation, Northampton, MA, USA. TEM images were obtained using a JEOL JEM2010 LaB6 (JEOL, Tokyo, Japan) operating at 200 keV. The samples were prepared on a lacy, carbon-coated, 400-mesh copper grid by adding each diluted nanoparticle solution dropwise. Filter paper was used to remove excess water, and the samples were dried under room conditions. The micrographs were processed using DigitalMicrograph software (3.57, CA, USA). The particle size distribution histogram was determined using ImageJ and OriginPro software (9, Northampton, MA, USA). XPS measurements were performed using a Thermo Scientific K-alpha XPS system (Thermo Scientific, Waltham, MA, USA) with a monochromatized Al Kα source operating with an energy of 1487 eV and a spot size of 400 µm. The binding energy (BE) scale was calibrated using the adventitious C 1s photoelectron peak at 284.8 eV. The core-level spectra were recorded with a pass energy of 50 eV and step size of 0.1 eV, while the spectra deconvolutions were carried out via CasaXPS software (http://www.casaxps.com). Before these measurements, each powder sample (as received) was pressed on double-sided carbon tape, and the MNPs@polyphenols–carbon tape systems were then degassed in a load-lock for 24h until a pressure greater than 1 × 10^−8^ Torr was achieved. Subsequently, the analysis was performed at a pressure close to 1 × 10^−9^ Torr and, at the same time, sample charging was compensated for with the help of an electron flood gun.

The topography and SMD analyses of the MNPs@polyphenols were performed by using a scanning probe microscope (SPM), JEOL-JSPM-5200, in the atomic force microscopy and magnetic force microscopy modes (AFM-MFM) or lift high mode. Each powder sample was confined to a carbon adhesive tape and flattened with the pressure of a flat glass. A magnetic tip NSC18, Co-Cr/Al *Mikromasch*, with an uncoated radius of 8 nm, a coated radius < 40 nm, and full-tip cone angle of 40°, was used for the MFM characterization. The magnetization of the tip was achieved using a neodymium magnet. Topography and MFM images were obtained at 75 kHz with a lift height interaction of 5–86 nm and outputs of 0.011–0.025 Amp/V and H = 5 kOe under saturation conditions. The topography and SMD images were processed and analyzed using Gwyddion software (2.61, Boston, MA, USA) for SPM.

## 5. Specific Loss Power (SLP) Measurements

The SLP quantifies the ability of MNPs to convert energy from an alternating magnetic field (AMF) into heat. For our samples of magnetic colloids with masses of MNPs ($m_{NP}$) dispersed in a liquid carrier $\left(m_{l}\right)$, the *SLP* was calculated using the following calorimetric relationship:(3)$$SLP=\frac{m_{NP}c_{NP}+m_{l}c_{l}}{m_{NP}}{\left(\frac{∆T}{∆t}\right)}_{max}$$
in which $c_{NP}$ and $c_{l}$ represents the specific heat capacities of the MNPs and the liquid carrier, respectively. For MNPs with a concentration lower than ≈7–9 mg/mL, the term $m_{NP}c_{NP}+m_{l}c_{l}$ can be approximated to $m_{l}c_{l}$. The expression in parentheses represents the heating rate of the system, which is evaluated at the maximum slope. The *SLP* values were obtained from derivation of the curve T(t), using the criterion of the maximum from the derivative *dT*/*dt* and employing the sample concentration as $\phi =m_{NP}/V_{l}$, where $V_{l}$ and $\delta_{l}$ are the volume and the density of the liquid carrier, as follows:(4)$$SLP=\frac{C_{l}\delta_{l}}{\phi }{\left(\frac{dT}{dt}\right)}_{max}$$

The term *dT*/*dt* was obtained from the first seconds of the initial temperature increase [57] to approximate the experimental conditions as a quasi-adiabatic environment [58]. The SLP values were measured using a commercial applicator (D5 from nB nanoscale Biomagnetics, Zaragoza, Spain). The experiments were conducted within a range of magnetic field intensities, with values ranging from 14.9 to 28.9 kA/m, at a fixed frequency of 764 kHz. The temperature was measured in situ using an optical fiber sensor. Milli-Q water was used as the liquid carrier during these experiments. Prior to each measurement, the concentration of the MNPs@polyphenol magnetic fluid was adjusted to approximately 5 mg/mL.

## 6. Biological Evaluation

### 6.1. K562 Cell Line

The human cell line K562 (chronic myelogenous leukemia, ATCC) was cultured in an RPMI medium supplemented with 10% FBS, 1% non-essential amino acids (100 mM), 1% L-glutamine (100 mM), and 1% antibiotic–antimycotic (100×) in a humidified chamber at 37 °C and 5% CO_2_. For cell viability assays, the cells were resuspended in a fresh medium, and 3 × 10^4^ cells were seeded into 96-well plates and incubated for 24 h.

### 6.2. Treatments

Stock solutions and serial dilutions of uncoated nanoparticles (NPsD), nanoparticles coated with *Punica granatum* extract (PG-NPs), nanoparticles coated with *Vaccinium corymbosum* extract (VC-NPs), or nanoparticles coated with *Vitis vinifera* extract (VV-Ps) at 0, 12.5, 25, 50, and 100 μg/mL were prepared in the RPMI medium using an ultrasonic processor (GEX 130 PB) for 30 s (130 watts, 20 kHz, with 40% ampl). Then, freshly prepared extracts of *P. granatum* (PG), *V. corymbosum* (VC), or *V. vinifera* (VV) were prepared in the RPMI, and serial dilutions were made at 0, 12.5, 25, 50, and 100 μl/mL except for the VC extract, which was in μg/mL.

### 6.3. Cell Viability Determined via MTT and NR Uptake (NRu) Assays

Serial dilutions of NPsD or PG–NPs, VC–NPs, or VV–NPs (previously sonicated) and the PG, VC, or VV extracts were immediately added to the cell cultures, and they were incubated for a total of 48 h. To determine the cell viability via an MTT biotransformation into formazan, 20 μL of MTT (5 mg/mL in PBS; 137 mM of NaCl, 2.7 mM of KCl, 10.1 mM of Na_2_HPO_4_, and 1.8 mM of KH2PO4 at a pH of 7.4) was added to the wells and incubated for 3 h. Then, the medium was discarded, and the plates were washed with PBS. The formazan crystals were dissolved with DMSO (100 μL/well), and the absorbance was determined using a spectrophotometer (Multiskan FC, Thermo Scientific) at 492 nm.

To determine the cell viability via the NRu assay, 25 μL of the NR solution (0.34 mg/mL in PBS; pH 6) was added to the wells and incubated for 3 h. Then, the medium was discarded, and cells were washed twice with PBS. Then, 100 μL of an acidic solution (water/ethanol/acetic acid at 49:50:1 *v*/*v*/*v*) was added to dissolve the NR inside the cells, and the absorbance at 540 nm was determined.

The values obtained from both assays were compared to the control value, which was considered 100% cell viability. The inhibitory concentration at 50% viability (IC_50_) was determined using the IC_50_ Calculator (AAT Bioquest, Inc., Pleasanton, CA, USA. Quest Graph™ IC50 Calculator. AAT Bioquest. https://www.aatbio.com/tools/ic50-calculator (accessed on 10 July 2023)).

The data are presented as the mean ± standard deviation of three independent experiments in triplicate. Using Student’s *t*-test, data were considered significant when *p* values were <0.05.

## 7. Results

### 7.1. Structural Analysis

The MNPs@polyphenols synthesized with organic extracts of Vitis vinifera, Vaccinium corymbosum, and Punica granatum were labelled as VV–NP (black line), VC–NP (blue line), and PG–NP (red line), respectively. The samples’ structure and their polyphenol capping were identified via XRD, Figure 1a.

The main diffraction planes corresponded to the FCC lattice of the magnetite (Fe_3_O_4_)’s inverse spinel cubic structure (JCPDS card no. 19-0629), with a lattice parameter of 8.396 Å and the space group Fd$\bar{3}$m (227). Herein, six intense characteristic-width peaks contributed, with higher X-ray scattering generated by small MNPs@polyphenols at the 2θ values of 30.19°, 35.45°, 43.23°, 53.72°, 57.39°, and 62.76°, corresponding to the crystallographic planes (220), (311), (400), (422), (511) and (440), respectively.

It can be noted that the VV–NP pattern has a weak diffraction peak at 53°, which could be due to the formation of an amorphous phase, capping the MNPs@polyphenols as a result of the partial oxidation of the Fe^2+^ ions and lattice distortion ($\epsilon_{VV}$ = 0.046835967, $\epsilon_{VC}$ = 0.04256, $\epsilon_{PG}$ = 0.042210834 for the VV–NPs, VC–NPs, and PG–NPs, respectively). On the other hand, the peak at 27.06° and the peak at 38.59°, which are identified with * in Figure 1a, could be due to the milling in the Punica granatum processing. The crystallite size of the synthesized MNPs@polyphenols <d> can be estimated using the Scherrer equation; it indicates a relationship between X-ray diffraction peak broadening, the size of the crystallite, and a profile function adjustment via the pseudo-Voigt function at FWHM = f(u,v,w).
(5)$$d=\frac{\;k\lambda }{\beta Cos\theta }$$
(6)$$\epsilon =\frac{\beta }{4}\tan\theta$$
where *d* is the average crystallite size, *k* = 0.95 is the shape constant, *λ* is the wavelength of CuKα, *β* is the full width at half-maximum (FWHM) of the most intense peak, and *θ* is the Bragg angle. The average size estimations for each sample were defined as 7.34 ± 0.06 nm, 8.09 ± 0.10 nm, and 8.15 ± 0.09 nm for the VV–NP, VC–NP, and PG–NP samples, respectively, and their lattice parameters are $a_{VV}$ = 8.368, $a_{VC}$ = 8.362, and $a_{PG}$ = 8.357 Å, respectively. This lattice distortion was identified from the small displacement of the diffracted peaks, which were associated with oxidation and small transformations from Fe_3_O_4_ to γ-Fe_2_O_3_ over the surfaces of the MNPs@polyphenols produced during the synthesis and the manipulation of the samples. However, the polyphenol coating can add scattering in the diffractograms, introducing an additional peak broadening.

An FTIR analysis was carried out to determine the bonding of the functional groups on the surfaces of the MNPs@polyphenols, also allowing for the determination of the iron oxide contributions. The IR spectra of the VV–NP and PG–NP samples (Figure 1b) showed a strong peak at about 576 cm^−1^ corresponding to the existence of vibrations due to stretching corresponding to the intrinsic nature of the Fe–O metal–oxygen band at the tetrahedral site. The presence of this band confirms that the present phase of the synthetized MNPs@polyphenols was Fe_3_O_4_ [59]. In comparison, the VC–NP spectrum showed a doublet at 576 and 658 cm^−1^, which correlated with the sequel of a possible oxidation on the surfaces of the MNPs@polyphenols.

On the other hand, the characteristic band identified at 3410 cm^−1^ results from the O−H stretching vibration, which can indicate hydrogen bonding between the polyphenols [60]. Following an FTIR analysis, in [61,62], the authors suggest that the –OH group was involved in the reduction of the Fe ions and the formation of the MNPs@polyphenols. In [63], they attribute the peak detected at 1625 cm^−1^ to phenolic hydroxyl groups (–OH); likewise, in [60], the authors associated the peak with the presence of polyphenols, which was expected as a consequence of the polyphenols coating the MNPs@polyphenols. According to [64], the 1632 cm^−1^ transmittance band is split into two close peaks corresponding to –OH and C=O groups. In addition, the transmittance peak observed at 1390 cm^–1^ could be a result of the C–OH stretching vibration [60]. So, this would originate from the polyphenol bonds on the surfaces of the MNPs@polyphenols, as would be expected from the presence of –OH bonds in the chemical structure of the extracts used. The transmittance band at 1029 cm^–1^ was attributed to the C–O acid stretching groups, corresponding to the range at 1000–1350 cm^–1^ [63]. This peak also was ascribed to alcohols, carboxylic acids, esters, and ethers [65].

In the VC–NPs, the modes at 803 cm^–1^ and 888 cm^–1^ could have originated from C–O–C vibrations, while the transmittance peak observed at about 803 cm^–1^ was ascribed to the C–C stretching mode [64,65]. Similar bands are reported for Vaccinium corymbosum [66]. As consequence of the FTIR results, the stretching vibrations of the polyphenols indicate that the extracts were successfully coated on the surfaces of the synthetized MNPs@polyphenols.

The particle size distribution and a morphology of a quasi-spherical shape were observed via TEM (Figure 2). Also, the polyphenol capping was defined as a small shell with a thickness of approximately 1.5 nm, surrounding the Fe_3_O_4_ nucleus. The particle size distribution histogram was determined via counting 100 MNPs@polyphenols. The mean particle sizes were 7.79 ± 1.63 nm, 8.66 ± 1.37 nm, and 8.8 ± 1.7 nm for the VV–NPs, VC–NPs, and PG–NPs, respectively (Table 2). Based on the Scherrer equation, the crystallite size of the synthesized product is in agreement with the results of smaller particle sizes obtained via the TEM. Similarly, the selected area electron diffraction (SAED) pattern revealed that the (220), (311), (400), (422), (511) and (440) diffracting planes and the interatomic distances observed on the selected area diffraction pattern are in agreement with structural XRD results, in which the well–defined diffraction rings correspond to the magnetite planes, while the polyphenol contribution was observed as scattered, overlapping rings.

The powder samples were investigated via X-ray photoelectron spectroscopy in order to analyze the surface chemical composition of the MNPs@polyphenols, including the oxygen–iron interaction and the effect of the polyphenols over the MNPs. The XPS core level and valence band signatures were analyzed using CasaXPS software (http://www.casaxps.com). All the spectra were corrected via Shirley background subtraction. Iron has been reported to exhibit complex multiplet splitting as a consequence of unpaired d electrons [67]. Accordingly, a multiplet structure for Fe 2p was taken into account; for instance, for the Fe 2p_3/2_ photoemission in Figure 3a, a spectral deconvolution was performed using seven doublets associated with different oxidation states (multiplets from 709 to 714 eV), high-binding-energy surface structures (715 eV), low-binding-energy “pre–peaks” (708 eV), and satellite peaks (718 eV). Similar considerations were used for the Fe 2p_1/2_ contributions. The pre-peaks were proposed to consider the formation of Fe ions with oxidation states of less than (2+) and (3+) due to the presence of defects in neighboring sites [68]. To carry out a proper deconvolution process, the peak-to-peak area ratio, doublet energy separation, and FWHM were considered, as reported elsewhere [13]. With respect to the peak-to-peak area ratio, for each Fe 2p doublet, a value close to 0.5 was found, which is in agreement with the (2J + 1) condition in which J is the quantum number for the total angular momentum [69]. The Fe 2p core-level photo emission peaks are shown in Figure 3a–c. For the VV–NP, VC–NP, and PG–NP samples, the lowest–BE Fe 2p_3/2_ and Fe 2p_1/2_ components of the multiplet were found at 710.50 and 724.10 eV, respectively [70]. The small peaks at 718.88 (VV–NPs, Figure 3a), 718.98 (VC–NPs, Figure 3b), and 719.19 eV (PG–NPs, Figure 3c) were related to a slight surface oxidation corresponding to a satellite peak of Fe 2p_3/2_ due to the presence of Fe^3+^ and Fe^2+^ [71]. It has been claimed that the γ-Fe_2_O_3_ phase displays visible satellite features, while these characteristic signals vanish for Fe_3_O_4_ [72]. In our case, the XPS and XRD results imply that both phases are mixed in the synthesized samples. In general, the positions of all the peaks were in good agreement with the previously reported results, with a small shift of ∆ = 0.20 eV toward higher BEs compared to the values reported for the magnetite peaks [68].

Shown in Figure 3d–f are the O 1s core-level spectra for the VV–NP, VC–NP, and PG–NP samples, respectively. For every sample, three peaks were found. The peak at the lowest binding energy remained almost constant at approximately 530.00 eV, with the signal of the PG–NP sample being centered at a slightly higher energy (530.13 eV). This peak has been related to metal–oxygen bonds (Fe–O) [73]. The second peak, related to Fe–OH bonds, appeared at 531.20 eV for the VV–NP and VC–NP samples and at a higher energy (532.47 eV) for the PG–NP sample [73]. The third peak exhibited changes in its position depending on the sample, i.e., for the VV–NP sample, it was found at 532.42 eV, while for the VC–NP sample, it was revealed at 532.49 eV, and it was found at 532.92 eV for the PG–NP sample. This peak has been associated with organic layers adsorbed on the samples (C–OH/C–O–C) [73]. The differences in the binding energies of the last two peaks can be associated with changes in the chemical environment due to the polyphenol capping.

The core-level C 1s spectra were deconvolved for the MNPs@polyphenols, as shown in Figure 3g–i. Three peaks were found for every spectrum. The peak at the lowest binding energies, i.e., 284.82, 284.97, and 284.84 eV for the VV–NP, VC–NP, and PG–NP samples, respectively, arose as a consequence of the C=C and C–H bonds formed due to the air exposure of the samples. Moreover, the middle peak, placed at 285.90 eV for the VV–NPs, 286.25 eV for the VC–NPs, and 286.27 eV for the PG–NPs, is in agreement with values reported in previous works for Fe_3_O_4_ nanoparticles covered with organic layers [13]. In addition, the signals placed at the highest energies were assigned to C–N, C–OH, or O–C=O bonds, which, respectively, appeared at 288.52, 289.03, and 288.47 eV for the VV–NP, VC–NP, and PG–NP samples.

Since some differences between the electronic structures of magnetite and maghemite were considered, a careful analysis of the valence band region allows this differentiation to be made. The XPS VB spectra are shown in Figure 4, in which these bands are formed by two peaks located at ca. 4.0 and 6.5 eV; these signals were, respectively, identified with the non-bonding (π) and bonding (σ) 2p orbitals of oxygen. In addition, the valence band maximum was obtained from a linear extrapolation of the low-energy region of the O 2p orbital. The determined values were 1.13, 1.29, and 1.43 eV for the VV–NP, VC–NP, and PG–NP samples, respectively. In all three samples, a small shoulder was observed around 1.5–0 eV, which was associated with an octahedral Fe^2+^ configuration present in the magnetite [72,74]. The elemental concentration, quantified via XPS, is presented in Table 3.

The topography was measured in atomic force microscopy (AFM) mode and, at the same time, magnetic force microscopy (MFM) was used to observe the interactions of the addressable magnetic domains [75], including their topology and particle size. As reference, the topography and profiles from the VC–NP sample were analyzed in Figure 5. First, the 2D topography image was obtained, and the MNPs@polyphenols can be observed as dispersion of the synthetized MNPs@polyphenols in which round agglomerates were formed via the pellet preparation, as shown in Figure 5a. The size of the nanoparticles in the VC–NP sample was estimated using the topography profile, as shown in Figure 5b, which indicates that the nanoparticle size was about 13 nm. The magnetization M_s_ of the bulk has a value of 92 emu/g, while ~55–73 emu/g at room temperature [48,65,65,76,77] was reported for functionalized MNPs of 13 nm. The organic coating could reduce the magnetization due to the non–magnetic nature of the extract and because the coating layer of polyphenols has a magnetization value of ~55 emu/g [78], depending on the thickness of the layer. According to superparamagnetic behavior, the expected MFM result was a demonstration of the coherent magnetization alignment of the MNPs@polyphenols as single–magnetic domains (SMDs).

These flux lines with a magnetic field were observed as a parallel array in lift mode for the VC–NPs. The MFM magnetic scanning conditions were applied at the initial conditions (H(0)) to reach magnetization in saturation state (H(↑)) and finally demagnetization (H(0). In Figure 6, 3D images show a sequence of topography and lift-mode phase images. Some differences were observed in the topology just before the saturation state; after that, small, round agglomerates were formed again from magnetic applied field.

Therefore, the MNPs@polyphenols possess the ability to return to their initial conditions after the application of a magnetic field. When the saturation condition (H(↑)) was reached, the magnetic moments were oriented in an addressable direction like the magnetic field and distributed in the surface of sample. Figure 6 shows the alignment of the magnetic field lines from single domains and their assembly. After that, in a demagnetized state (H(0)), there was a random distribution of SMDs, but a few aligned remanents were maintained. According to the MFM images and the profiles obtained under the variation in the distance between the tip and sample, the magnetic domains have uniaxial anisotropy.

The zoom of the saturation state (Figure 7a,b) shows two regions in which the magnetic domains were under the action of the magnetic field at H = 5 kOe. As a result, the SMDs were oriented at 90°. The profile (Figure 7c) allows us to observe a uniaxial behavior. The flux lines over the magnetic domains were around 2.7 nm.

### 7.2. Cell Viability via the MTT and NRu Tests

We evaluated the cytotoxicity of the natural extracts and the nanoparticles on K562 cells, using the MTT and NRu assays. The cellular metabolic activity of mitochondria (MTT assay) (Figure 8) shows that the NPsD do not affect the viability of the K562 cell line, the VC extract significantly reduces viability in a dose–dependent manner, decreasing cell viability from a concentration of 12.6 μg/mL (55%) to 100 μg/mL (20%), and the PG extract also decreases the cell viability depending on the dose, from a concentration of 12.5 μg/mL (90%). The IC_50_ were 24.8 and 59.0 μg/mL for the VC and PG extracts, respectively, and the VV extract did not modify the viability at of the different concentrations evaluated. While when the covered NPsD were evaluated, the VC, PG and VV–NPs did not modify cell viability at any concentrations.

The NRu assay, which indicates the integrity of the cell membrane and accumulates in lysosomes [79], showed that the NPsD reduced cell viability by 20% only at the concentration of 12.5 μg/mL when compared against the control group. The VC and PG extracts reduced cell viability from 12.5 ug/mL (35 and 58% respectively) and to 5 and 10% from a concentration of 50 y 100 μg/mL, with a dose-dependent manner observed in the PG extract. The VV extract reduced cell viability from a concentration of 50 μg/mL (80%), and the IC_50_ values were 20.7, 51.9, and 38 μL/mL for the VC, PG, and VV extracts, respectively. In a similar way as to what was observed in the MTT assay, when the extracts cover the NPsD, the VC, PG and VV–NPs do not affect the cell viability the of K562 cell line [80,81].

The IC_50_ (the half-maximal inhibitory concentration) was determined for all extracts with IC_50_ values of 24.8, 59 and 32.9 μg/mL of MTT and 20.7, 51.9 and 38 μL/mL of NRu of *P. granatum*, *V. vinifera* and V. *corymbosum*, respectively.

### 7.3. Power Loss under AC Magnetic Fields

The observed inductive heating response showed the expected increase in the SLP values with an increase in the applied field amplitude $H_{0}$ (Figure 9), with the highest SLP values obtained for the VC–NPs (187.16 W/g) and the VV–NPs (121.83 W/g) at the maximum field ($H_{0}$ = 28.9 kA/m), while the PG–NPs showed lower values (73.03 W/g). The heating capacity is known to depend on the MNP’s particle size, shape, and magnetic anisotropy. Previous reports found SLP values of up to 203 W/g in γ-Fe_2_O_3_ MNPs of similar physical parameters, consistent with the values in the present work. Also, reports on Fe_3_O_4_ MNPs functionalized with glutamic acid (*d*~9 nm) found SLP values of 130 W/g ($H_{0}=$ 35.8 kA m^–1^, f= 316 kHz) [77].

## 8. Discussion

It is interesting to note the partial surface oxidation in our MNPs@polyphenols coated with polyphenols, which can be ascribed to the oxygen sensibility surface from the iron oxide with a small nanometer size, as suggested by the small displacements in the main peaks from the XRD data. Similarly, the XPS spectroscopy data at the Fe 2p edge showed a small satellite peak at each sample (Figure 3), suggesting the presence of a maghemite phase. This is consistent with a tiny layer of maghemite on the surfaces of the MNPs, possibly from the protocols carried out during the synthesis process. Superficial oxidation was observed in similar works with the synthesis of nanoparticles using similar organics [23]. Therefore, the data suggest that our samples have a Fe_3_O_4_@γ-Fe_2_O_3_ core–shell structure comprising a magnetite core with a thin maghemite capping. The oxidation of the polyphenols accelerates the crystal nucleation and growth, reducing the overall size in the process [82]. Fe^2+^ strongly stabilizes due to the poylphenols’ ligands but rapidly oxidizes in the presence of oxygen to produce Fe^3+^–polyphenol complexes; this phenomenon commonly known as autoxidation [83]. However, the surface oxidation of the MNPs could not be inhibited in our case. Some of the features of the polyphenols and the coating thickness are the main parameters to be considered in future studies. The oxidation of Fe_3_O_4_ was analyzed via XRD as a function of time. The lattice parameters changed to 8.385, 8.386, and 8.370 Å for the VV–NPs, VC–NPs, and PG–NPs, and the MNPs turned brown.

The vibrational mode of the Fe−O bond from the FTIR measurements could be referred to as the main characteristic of the magnetite phase. This peak in the spectra of the PG–NPs was more structured and intense compared to the other samples, probably originated in the phenolic components within the extracts of *Punica granatum* that affect the cation distribution in the tetrahedral and octahedral sub-lattice [84]. In addition, functional groups in the organic layer, which stabilize the obtained MNPs@polyphenols, were identified. Bonds of C–O and OH–were spotted, as was expected due the polyphenols on the surfaces of the MNPs@polyphenols. The latter bond is important because of its capacity to bind heavy metals [85,86].

Polyphenols have been used as stabilizer agents because this action is useful for the stabilization of nanoparticles. Also, polyphenols have the capacity to act like phyto-reductants, therefore enhancing the receptiveness of the nanoparticles for cellular uptake. In the biomedical application of nanoparticles, their stability and transmittance can be enhanced by the coatings.

Phenolic compounds are useful for encapsulating nanoparticles; in addition, they can improve their bioavailability [87]. Metal ions can interact with polyphenols; the ions are then are reduced to a nanometric size, then led to the formation of nanoparticles. This indicates that ${Fe}^{3+}$ can be reduced to ${Fe}^{2+}$ by an organic extract. During synthesis, the polyphenols were linked to the MNPs@polyphenols through Fe–OH bonds, which can be in the form of ${Fe}^{2+}$–O or ${Fe}^{3+}$–O, a covalent bond [88]. Actually, some flavonoids prefer metal bindings at the 3-hydroxyl-4-carbonyl group, 4-carbonyl-5-hydroxyl group, and 30–40 hydroxyl sites [89]. However, the ability of polyphenols to form both covalent and noncovalent bonds depends on their phenolic compound size, and a thicker coating layer is the outcome of a greater amount of polyphenols adsorbed [90].

In the past, there were some attempts to deconvolve the high-resolution XPS spectrum of Fe 2p with the goal of identifying different iron oxide phases, e.g., magnetite and maghemite [91]. Nevertheless, it should be mentioned that these analyses were strongly based on specific criteria assumed by the authors. For instance, Biesinger et al. [1] compared peak shapes to the theoretically calculated multiplet split peak shapes obtained by Gupta and Sen [92], claiming that they found a relatively good agreement. In this work, Biesinger et al. considered up to thirteen components (one peak for Fe^0^, five peaks for Fe_2_O_3_, and seven peaks for Fe_3_O_4_) in the Fe 2p_3/2_ signal, without taking into account satellite components. Pratt et al. [93] analyzed Fe_2_O_3_, considering five peaks to deconvolve the Fe 2p_3/2_ signal. On the other hand, Lin et al. [94] used the shake-up satellite positions as guides for the positioning of the most intense Fe 2p peaks and the subsequent quantification of the Fe^0^, Fe^2+^, and Fe^3+^ components in a series of thin oxide films. From these examples, it can be emphasized that any effort made to fit two or more components into a Fe 2p spectrum will involve component overlapping and lead to an intrinsic fitting error. In this sense, the reliable identification of iron oxide phases still remains as an open scientific problem. The surface elemental concentration was determined via XPS (as presented in Table 3) [95,96]. Even though different extracts were used for the synthesis of the NPs, the VV–NP and VC–NP samples exhibited almost the same contents of iron, oxygen, and carbon. Among them, the VV–NP and VC–NP samples presented higher amounts of Fe and O, which could be related to higher Fe_3_O_4_ or γ-Fe_2_O_3_ contents on the surfaces of these samples. On the other hand, a higher quantity of polyphenols is expected for the PG–NP sample, in accordance with the greater presence of carbon found on its surface.

The TEM size analysis shows that the MNPs@polyphenols have a quasi-spherical shape and mean sizes of 7.7, 8.6, and 8.8 nm for *V. vinifera*, *P. granatum*, and *Vaccinium C*, respectively (Table 2). The smaller sizes of the VV–NPs and PG–NPs could be the result of the presence of high-molecular-weight polyphenols in the aqueous extract [97]. There was no difference in the size distribution of the VC–NPs and PG–NPs, but the VV sample has a wider size distribution than the other two. Consequently, in magnetic effects, the larger particles will dominate [98]. According to the XRD and SAED analyses, the MNPs@polyphenols have a crystalline structure, and the main plane of formation was (111). The extract coating of a thickness of 1.5 nm was observed as a thin film in HRTEM, showing a core–shell nanostructure. An analysis of the magnetic behavior of the VC–NPs via MFM demonstrated that they can be addressed by a magnetic field. Additionally, 3D images show that the magnetic domains have the structure of single magnetic domains with a uniaxial response.

The MNPs@polyphenols demonstrated the ability to be addressed by a magnetic field, and owing to their chemical properties, they can be candidates for possible biomedical applications such as magnetic hyperthermia and targeted drug delivery.

The natural extracts showed reductions in cell viability for the PG and VC extracts as the function of concentration was increased. This was observed at a major concentration of 100 µg/mL. However, the uncoated nanoparticles (NPsD) did not affect cell viability, and the PG–NP, VC–NP, and VV–NP samples showed similar responses, with minimum effects in a K562 cell line, because the concentration of the polyphenols was diminished to small quantities in the MNPs@polyphenols. From the thicknesses of the PG–NPs and VC–NPs, the polyphenol concentrations were estimated to be 16.11 and 15.96%, respectively. The results showed that SLP values are influenced by the average particle size and the magnetic field and frequency. The highest values were obtained for the VC–NPs and VV–NPs (187.16 and 121.83 W/g, respectively). The SLP values observed in this study were higher than those previously reported for functionalized MNPs and can be attributed to the use of a higher frequency during the magnetic stimulation, indicating that they could be useful as drug delivery systems.

## 9. Conclusions

We have synthetized MNPs@polyphenols via a modified co-precipitation method, using the extracts of *Vitis vinifera*, *Punica granatum*, or *Vaccinium corymbosum*. This direct, easy co-precipitation method is cheaper and more efficient, and the aqueous extract can act as a reducer and stabilizer, as well as the capping agent. The MNPs have a crystalline structure with particle sizes of 7.3 nm, 8.1 nm, and 8.2 nm for the VV–NPs, PG–NPs, and VC–NPs, respectively. The structure of the MNPs@polyphenols was matched to a magnetite phase. A small lattice distortion was identified and associated with surface oxidation over the MNPs@polyphenols during the synthesis process. The MNPs@polyphenols were observed via TEM, and the thicknesses of the amorphous layers were estimated in a range of 1.5 nm; this contrast was identified according to the functional groups of the extract. Also, the presence of a Fe–OH bond was corroborated, which was a sign of linking between the MNPs@polyphenols and the organic bonds –OH, C–O, C–C, O–C=O, and C–OH.

Polyphenols possess the ability to have strong bonding affinities via the following types of bonding: noncovalent via multiple-hydrogen bonding, hydrophobic, electrostatic, hydrophobic, π–π and cation–π interactions, the van der Waals force, and covalent. Consequently, polyphenols possess the ability to bind with molecules. Owing to the anticancer, anti-metastatic, antibacterial, antiviral, and anti-inflammatory activities of polyphenols, they are used in biomedical applications. In addition, the cellular uptake of nanoparticles through endocytosis could be enhanced by the action of natural polyphenols such as epigallocatechin gallate, which is present in *Punica granatum* and *Vaccinium corymbosum*. But this is a challenge to be explored in detail to determine their possible applications.

The elaborated MNPs@polyphenols represent an option to be considered as localized inductive heating agents in a liquid carrier. The observed SLP values demonstrate the potential for therapeutic magnetic hyperthermia, with enhanced biocompatibility achieved via the polyphenol coating.

The SMD interactions demonstrated ensure the presence of Néel contributions in the relaxation mechanisms. For this reason, the MNPs@polyphenols are suitable for inductive heating and energy dissipation, even in intracellular space.

## Data Availability

Not applicable.

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
