# Peer review of "Direct Polyphenol Attachment on the Surfaces of Magnetite Nanoparticles, Using Vitis vinifera, Vaccinium corymbosum, or Punica granatum"

_nanomaterials, 2023, doi:10.3390/nano13172450_

Round 1
Reviewer 1 Report
The manuscript “Direct polyphenol attachment on magnetite nanoparticles surface with Vitis vinifera, Vaccinium corymbosum or Punica granatum” by Matias-Reyes et al. is a well-written manuscript on the synthesis of magnetic nanoparticles modified with polyphenols.
The manuscript is nicely introduced and discussed.
There are a few minor issues which need to be addressed:
Abstract
Introduction
It is unclear if MNP refers to magnetic nanoparticles or magnetite nanoparticles since both descriptions are used in the text. Please either just use one explanation or use two different abbreviations and do not use it as synonyms.
Experimental:
“The synthesis was performed following the co−precipitation method [49] with some modifications [50,51]. The phytochemicals methods followed has been consider eco−friendly, cheaper and efficient [52].”
Please exactly state all modifications. It is unclear why you reference four papers for one synthesis.
Please explain what the phytochemicals methods are.
Formation mechanism
“This is generated through the chelation of Fe3+−OH and Fe2+−OH; or in other cases with Fe3+−COOH and Fe2+−COOH on the surface of Fe3O4 nanoparticles [55].”
The reference describes no chelation through OH bonds. It is indeed highly unlikely that you will establish covalent bonds through the hydroxyl groups of polyphenols with iron ions on the iron oxide particle surface. I also criticize the reference 55 which is not a very good evidence for a complexation on the surface since there is not a discussion about the shift in the IR vibrations but just the existence of C=O vibrations is used as an evidence which is not correct.
“The crystallite size of synthesized MNPs@polyphenols can be estimated using the Debye−Scherrer equation, it indicates a relationship between X−ray diffraction peak broadening and crystallite size and profile function ajustment at PseudoVoigt at FWHM=f(u,v,w).”
The equation is named solely after Paul Scherrer. Please refer to it a Scherrer equation and not Debye-Scherrer equation.
Also I doubt the accuracy of this method. Please do not indicate an accuracy in the range of picometer.
I also doubt that the XRD diffractograms are suitable to differentiate between magnetite and maghemite. You have different polyphenols and small nanoparticles which show a peak broadening due to their lattice errors. Therefore, it is not possible to distinguish between the gamma-Fe2O3 phase and Fe3O4.
All in all I recommend minor revisions.
Reviewer 2 Report
The manuscript "Direct polyphenol attachment on magnetite nanoparticles surface with Vitis vinifera, Vaccinium corymbosum or Punica granatum" describes the synthesis of magnetic nanoparticles coated with polyphenols from specific plants. The work is interesting with many details and I propose to publish it with minor corrections concerning the following points:
1. Although their characteristics are described in detail in the introduction, I ask the authors to describe the reasons for choosing exactly these plant species. (similar or very different chemical composition, acidity of the extract, targeted therapeutic effect, or other).
2. The (111) peak is not given in the X-ray patterns why?
3. In the X-ray pattern of PG-NP, there are several weak, narrow peaks that do not belong to magnetite. I suppose they are of quartz or other silicate material, but I ask the authors to note this in the text.
4. Determination of the average crystallite size is accurate to the first decimal place. Please correct your "too accurate" data. (e.g. size 7.3±0.1 nm). On the other hand, the authors did not provide the error in determining the unit cell parameter of the obtained magnetite materials. Please add it.
5. From the given data for the unit cell parameter, it is clear that the magnetite is partially oxidized. Which is normal for a nanosized material. Does the presence of polyphenols on the surface delay magnetite oxidation or accelerate it?
6. About FTIR analyses. The band at 576 cm-1 is associated with Fe-O vibrations in the tetrahedron. The octahedron band is typically located around and below 400 cm-1. Please correct the text.
7. What specific compounds do you think are responsible for the extra bands in the FTIR spectrum of VC-NP sample?

Reviewer 3 Report
Matias-Reyes et al. synthesized MNPs@polyphenol by relatively easy method. They characterized the MNPs using PXRD, IR, XPS, AFM and so on. They also investigated the cell activity for the MNPs. The results and the discussion are reasonable. The present referee thinks the present manuscript should be accepted. But before that the present referee has small comments.
Using Debye-Scherrer equation, the average size is estimated. The particle size is also estimated using TEM. The sizes obtained from the different method are almost the same for VV-NP and VC-NP. On the other hand, the size is a little bit different (8.15 and 12.91) for PG-NP. Is there any reason?
Using the amount of carbon in Table 2, the relative ratio between iron oxide and polyphenol is discussed. Is it possible to discuss the relative ratio between magnetite and maghemite by using the relative ratio between Fe and O in Table 2?
Cell viability decreases with increasing concentration for PG and VC, while the cell viability is almost the same independent of the concentration for PG-NP and VC-NP. What is the difference between PG, VC and PG-NP, VC-NP?
